# Statistical ensembles without typicality

Paul Boes[1], Henrik Wilming[1], Jens Eisert[1] & Rodrigo Gallego[1]

Maximum-entropy ensembles are key primitives in statistical mechanics. Several approaches have been developed in order to justify the use of these ensembles in statistical descriptions. However, there is still no full consensus on the precise reasoning justifying the use of such ensembles. In this work, we provide an approach to derive maximum-entropy ensembles, taking a strictly operational perspective. We investigate the set of possible transitions that a system can undergo together with an environment, when one only has partial information about the system and its environment. The set of these transitions encodes thermodynamic laws and limitations on thermodynamic tasks as particular cases. Our main result is that the possible transitions are exactly those that are possible if both system and environment are assigned the maximum-entropy state compatible with the partial information. This justifies the overwhelming success of such ensembles and provides a derivation independent of typicality or information-theoretic measures.

[1] Dahlem Center for Complex Quantum Systems, Freie Universität Berlin, 14195 Berlin, Germany. Correspondence and requests for materials should be addressed to P.B. (email: pboes@zedat.fu-berlin.de)

Maximum-entropy ensembles, such as the micro-canonical or the canonical ensemble, are the pillars on which statistical mechanics rests. Given some partial information about a system, a vast set of predictions about its behaviour can be derived by assigning to the system that statistical ensemble which maximises the entropy compatible with the partial information. Yet, in some ways this assignment may be seen as being peculiar in that there exist many other possible physical states that are compatible with this information. The assignment of maximum-entropy ensembles is primarily justified by its undoubtable empirical success when it comes to an agreement with experiment and observation. Thus, unsurprisingly, there has been much work aiming at providing theoretical grounds which explain its empirical success, going back to seminal work by Gibbs[1]. The most successful general arguments justifying the use of ensembles—both for classical and quantum systems—are either based on specific assumptions of the microscopic interactions from which ergodicity can be derived (see refs. [2,3] for a review on this approach and its conceptual problems), or based on the notion of typicality. The latter is the observation that the volume of pure quantum states (compatible with the information) that behave like a maximum-entropy ensemble is close to unity, with respect to a relevant measure on state space[4–6]. In these approaches, partially motivated by efforts in quantum thermodynamics[7,8], the aim is to show that the system at hand behaves like the ensemble in the precise sense that it will output the same measurement statistics for a restricted, but most realistic and relevant, set of observables. In this way, the agreement between experiments and the assignment of ensembles is justified, with the only notorious problem that the measure that produces the typicality is difficult to justify dynamically. There have been attempts to derive precisely the emergence of canonical ensembles for most times from microscopic dynamical laws for common locally interacting quantum systems (for reviews, see refs. [9–11]). However, it seems fair to say that it is still not fully clear yet why the probability of a system being, at any (or most) times, in a state should be described by this measure—a state of affairs particularly significant in light of the importance of this ensemble.

In this work, we provide a very different justification for the use of such ensembles. In contrast to the approaches mentioned before, our aim is not to derive that system's measurement statistics mimic those of the ensemble. Instead, we look at the possible state transitions that can be induced on a system from which one has only partial information (see also ref. [12]). More precisely, we consider an initial system described only by partial information in the form of the expectation value of a set of observables. We pose the problem of finding the set of transitions that this initial system can undergo by evolving jointly with an environment when the state of this environment is itself known only partially, that is, up to expectation values with respect to a set of observables that correspond to those of the system. The environment plays the role of a usual heat bath and the set of transitions encode any possible task: extracting work, reaching a colder/warmer state, performing a computation or any other. Our main result is that, for any initial state, the possible state transitions on such a system under partial information coincide exactly with those possible if the system and the environment were initially in the maximum-entropy ensemble state compatible with the partial information. This then not only justifies the use of the canonical ensemble to represent a system under partial information, it also allows one to derive the building blocks of phenomenological thermodynamics without assuming systems to be represented by this ensemble. In fact our results can be seen as a derivation of the Gibbs entropy and the Clausius inequality without a priori assigning equilibrium states to the systems

involved. Finally, since our results hold for any initial state, they do not suffer from the problem of typicality approaches mentioned above and allow us to avoid assumptions about the system's Hilbert-space dimension (apart from being finite). In particular, our results also hold for small, individual quantum systems.

## Results

**Motivating example.** We begin the presentation of our setting with a motivating example. Consider a small quantum system S with Hamiltonian $H$ within an environment E at temperature $T$ and with Hamiltonian $H_E$, that is, an environment in the canonical ensemble at that temperature and Hamilonian.

Given an initial quantum state $\rho$ of the system, we can ask which final states of the system can be reached by coupling the system to the environment and evolving the joint system SE in such a way that the global entropy and energy remain unchanged, if one assumes perfect control over both the environment Hamiltonian $H_E$ and the coupling, but for a fixed temperature $T$. Naturally, the answer to this question will strongly depend on the particular initial quantum state of S. For instance, the maximally mixed state $\rho = \mathbb{I}_S$ and an energy eigenstate $\rho' = |E_i\rangle\langle E_i|$ will generally allow for very different state transitions. That is, there will exist some final state $\rho_f$ that can be reached by some entropy and energy preserving procedure $O$ from $\rho'$, while no such procedure exists for $\rho$. Call this scenario the microstate scenario, because here one has full information about the actual 'microstates'—i.e. quantum states—of the system and the environment.

Suppose now that, instead of knowing the exact state of the system, one initially only knows its mean energy to be $e$ with respect to $H$. We capture this partial information in what we call a macrostate of the form $(e, H)$. In this case, one can again ask which are the reachable states given that partial information. However, in this case the difficulty is that, in general, there will be many microstates compatible with this information. For instance, suppose that $(e, H)$ is compatible with both $\rho$ and $\rho'$. In this case $\rho_f$ cannot be reached anymore because there is at least one state— $\rho$ in the previous example—compatible with the initial information for which $\rho_f$ is unattainable. That said, one concludes that in order to reach some final state $\rho_f$, if only partial information about the initial state of S is had, one requires a single operational procedure $O$ that takes any state compatible with the initial information to $\rho_f$. Note that this scenario is undesirably asymmetric in that the system's state is represented by a macrostate $(e, H)$ (capturing our partial knowledge), while the environment microstate is fully known to be in the canonical ensemble at temperature $T$. Hence, one can go one step further and consider a situation in which not only does one only know the system's initial mean energy, but also the environment is described by a macrostate $(e_E, H_E)$. In this case, it becomes even more difficult to reach a given final microstate $\rho_f$, since now there has to exist a single procedure $O$ that prepares $\rho_f$ from any microstate of S compatible with $e$ and any environment microstate compatible with $e_E$. Indeed, it may seem that in general no transition is possible under these circumstances. At the same time, this scenario most accurately describes the situation that one in fact faces in phenomenological thermodynamics, where only coarse-grained information is had about both system and environment. Call this last scenario then the macrostate scenario, because here both system and environment are described by macrostates $(e, H)$ and $(e_E, H_E)$, respectively.

The main result of this work is to show that, not only do there exist possible transitions in the macrostate scenario, moreover these transitions are fully characterised by assigning maximum-

entropy ensembles to the macrostates involved: Under a natural model of operational procedures modelling thermodynamic transitions that we introduce below, given some value $e$, a final microstate $\rho_f$ can be reached in the macrostate scenario if and only if it can be reached in the microstate scenario from the canonical ensemble state of energy $e$. Since the canonical ensemble is moreover the only state for which this equivalence holds, this result provides an explanation for the important role that the canonical ensemble plays in statistical mechanics, a theory formulated in the microstate scenario, to describe phenomenological thermodynamics, a theory formulated in the macrostate scenario.

**Formal setting**. We now proceed to make the notion of the microstate- and macrostate scenario rigorous and introduce our model of thermodynamic transitions, i.e. the transitions that a system S can undergo together with an arbitrary environment at fixed temperature.

Consider a $d$-dimensional quantum system S whose mean energy with respect to the Hamiltonian $H$ is known to be $e$. We refer to the pair $(e, H)$ as the 'macrostate' of the system, as it corresponds to a state of coarse-grained information about the system. Note, however, that we do not assume that the system is macroscopic, i.e. that $d \gg 1$. Every macrostate of the system corresponds to an equivalence class $[e]_H$ of 'microstates' $\rho \in D(\mathcal{H})$ of the system, namely all those density matrices whose mean energy with respect to $H$ is $e$, with $\mathcal{E}(\rho) := \text{tr}(\rho H) = e$. The canonical ensemble corresponding to a macrostate $(e, H)$ is then

$$\gamma_e(H) := \frac{e^{-\beta_S(e)H}}{\text{tr}(e^{-\beta_S(e)H})}, \tag{1}$$

where $\beta_S(e)$ is chosen such that $\text{tr}(\gamma_e(H)H) = e$. Note that, by construction, $\gamma_e$ is the maximum-entropy element in $[e]_H$ and exists for every macrostate. As is clear from the example, in the following, we will often be concerned with making comparative statements about the microstate- and the macrostate scenarios. To simplify the presentation and highlight similarities between these scenarios, we now introduce the following convention: Let $M$ be any map acting on microstates. Then $M((e, H)) := M([e]_H)$ is the corresponding macrostate-level map. This notation will prove convenient in several ways. For instance, the requirement that an operation $O$ maps all the states $\rho$ compatible with $(e, H)$ into the state $\rho_f$ is simply expressed by

$$O((e, H)) = \rho_f. \tag{2}$$

Similarly, this notation can be also used to express operations on tensor products of macrostates. For instance, the expression

$$O((e, H) \otimes (e_E, H_E)) = \rho_f \tag{3}$$

implies that $O(\rho \otimes \rho_E) = \rho_f$ for all $\rho$ and $\rho_E$ compatible with $(e, H)$ and $(e_E, H_E)$ respectively.

**Thermodynamic operations on macrostates**. Let us now describe and justify more precisely the form of a general macrostate operation as informally described above. With these operations we aim at capturing in full generality any possible transition that a system can undergo together with a heat bath. Hence, in order to describe an arbitrary macrostate operation, one is perfectly free to choose as an environment any system of arbitrary Hilbert-space dimension and with an arbitrary Hamiltonian $H_E$. As mentioned before, we do not assume that E is in a canonical ensemble—which would be fully determined by the inverse temperature $\beta := (k_B T)^{-1}$, dimension, and Hamiltonian—but to have a partial description in terms of its average energy,

thus assigning to it a macrostate $(e_E, H_E)$. We assume, as it is standard when considering thermodynamic operations[13–15], that the system and the environment are initially uncorrelated, hence one initially possesses the macrostate compound $(e, H) \otimes (e_E, H_E)$. Naturally, the attachment of an uncorrelated environment can be iterated an arbitrary number of times, say $N$, bringing each time a new environment with an arbitrary dimension and Hamiltonian.

Moreover, since the macrostates provided by the environment model a bath, it is natural to assume that there exists a functional relationship between the environment Hamiltonian and the energy. In particular, we will assume this relationship to be that $e_E = e_\beta(H_E)$, where

$$e_\beta(H_E) := \text{tr}\left(\frac{e^{-\beta H_E}}{\text{tr}(e^{-\beta H_E})} H_E\right) \tag{4}$$

is the thermal energy of a bath at inverse temperature $\beta$ and with Hamiltonian $H_E$. This assumption will be further discussed below. Dropping further the dependence on the Hamiltonian in (4) when it is clear from the context, the most general form of an initial macrostate then is of the form

$$(e, H) \overset{N}{\underset{i=1}{\otimes}} (e_\beta, H_{E^i}). \tag{5}$$

Given this model of the environment, we now turn to the describing the model of the joint evolution. Here, we aim at modelling the isolated evolution of SE, in the sense that it preserves the energy and entropy of the compound. Regarding the energy, one has to take into account that only mean values of the energy are accessible, hence it is most reasonable to impose only that the mean energy is preserved[16–18], while noting that the mean energy must be preserved for all the initial microstates compatible with our initial macrostate (5). Regarding entropy conservation, we enforce it by imposing a unitary evolution of the compound. We note, however, that our results also hold for larger set of operations such as probabilistic mixtures of unitaries or entropy non-decreasing operations, or even more generally, any set of operations that contains unitary evolutions as a particular case.

Let us now, for sake of clarity, enumerate the assumptions that come into play when describing macrostate operations:

Assumption 1: (Thermal energy environments) Given an environment with Hamiltonian $H_E$, then the associated macrostate is given by $(e_\beta(H_E), H_E)$, where $e_\beta(H_E)$ is the thermal energy at reference temperature $T$.

Assumption 2: (Uncorrelated subsystems) One can incorporate environmental systems that are initially uncorrelated with the initial system.

Assumption 3: (Unitary evolution) The compound SE undergoes a unitary evolution.

Assumption 4: (Global mean energy conservation) The unitary evolution of SE is such its mean energy is preserved for all the states (both of S and E) compatible with our partial information.

Before turning to the formal definition of macrostate operations on the basis of these assumption, let us briefly comment on the assumption that environment macrostates have thermal energy (4). Clearly, this amounts to assume that environment macrostates have the same mean energy as the

canonical ensemble at inverse temperature $\beta > 0$,

$$\gamma_\beta(H_E) := \frac{e^{-\beta H_E}}{\text{tr}(e^{-\beta H_E})}, \quad (6)$$

where we make the convenient abuse of notation of writing $\beta$ directly as the subindex, unlike (1) where the mean energy was used instead. This is indeed unproblematic since $e$ and $\beta$ are in one to one correspondence, hence we will use $\beta$ or $e$ indistinctively when it is clear from the context.

We emphasise that (4) does not amount to assuming that the environment is in the canonical ensemble—which would beg the question by giving a prominent role to the canonical ensemble—since many states other than the canonical ensemble fulfilling (4) exist. Nevertheless, Assumption 1 could raise the criticism that our further results—the justification of ensembles—rely on a seemingly arbitrary energy assignment for the macrostate of E, as given by (4). However, we show in the Supplementary Methods 1 that (4) is the only possible assignment so that macrostate operations reflect indispensable features of thermodynamical operations. More precisely, we prove that (4) is the only energy function that does not allow one to extract an arbitrary amount of work from E alone—even if only partial information is given. Even more dramatically, it is the only energy function that does not trivialise macrostate operations, in the sense that any possible transition would be possible. Hence, (4) can be regarded as a necessary feature of an environment so that thermodynamic operations are sensibly accounted for in the formalism.

Finally, combining the notational convention for operations on macrostates, Assumptions 1–4, and denoting the global mean energy as $\mathcal{E}(\rho_{SE}) := \text{tr}(\rho_{SE} H_{SE})$, we define formally the set of macrostate operations with an environment at inverse temperature $\beta$:

Definition 1: (Macrostate operations) We say that $\rho_f$ can be reached by macrostate operations from $(e, H)$, which we denote by

$$(e, H) \xrightarrow{\beta\text{-mac}} \rho_f, \quad (7)$$

if for any $\epsilon > 0$ and $\epsilon' > 0$ there exists an environment—that is, a set of $N$ systems with respective Hamiltonians $H_{E^i}$—and a unitary

$U$ on SE, so that

$$\rho_f \approx_\epsilon \text{tr}_E\left(U(e, H)\overset{N}{\underset{i=1}{\bigotimes}}(e_\beta, H_{E^i})U^\dagger\right) \quad (8)$$

while preserving the overall mean energy

$$\mathcal{E}\left(U(e, H)\overset{N}{\underset{i=1}{\bigotimes}}(e_\beta, H_{E^i})U^\dagger\right) \approx_{\epsilon'} \mathcal{E}\left((e, H)\overset{N}{\underset{i=1}{\bigotimes}}(e_\beta, H_{E^i})\right). \quad (9)$$

Here, we use $\approx_\epsilon$ to say that two quantities differ by at most $\epsilon$ in trace-norm, or in absolute value for expectation values. Note that although we allow for errors $\epsilon$, $\epsilon'$ in the transition and in the mean energy conservation, those errors can be made arbitrarily small, hence it is for all practical purposes indistinguishable from an exact transition with exact mean energy conservation. It is also important to stress again that, in the previous definition and following the notation introduced with Eq. (3), both (8) and (9) have to be fulfilled for all the microstates compatible with the macrostates appearing in those equations. See Fig. 1a for a schematic description of macrostate operations as presented in Definition 1.

**Thermodynamic operations on microstates and main result**. As stated before, our main result consists in showing that not only is the set of reachable microstates under macrostate operations in general non-empty, it can also be characterised exactly by the corresponding canonical ensembles. In order to be able to state this correspondence between macrostates and their canonical ensembles formally, we will now introduce microstate operations as the corresponding model of thermodynamic transitions in the microstate scenario. These differ from macrostate operations only in that we assign a particular microstate to S and E. In other words, microstate operations are the complete analogue of the operations in Definition 1, but with full information about the actual quantum states involved. Hence, conditions (8) and (9) are modified, for microstate operations, in that they have to be fulfilled for a single state and not for a set of states compatible with our knowledge.

Definition 2: (Microstate operations) We say that $\rho_f$ can be reached by microstate operations from $\rho$, which we denote by

$$\rho \xrightarrow{\beta\text{-mic}} \rho_f, \quad (10)$$

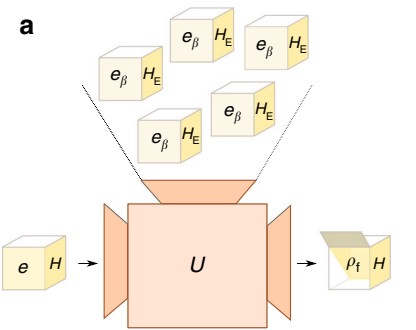
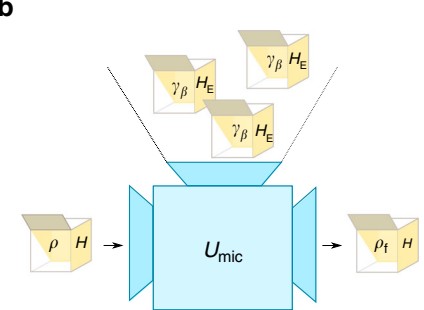

**Fig. 1** Pictorial representation of the equivalence between macrostate operations and microstate operations. Panel **a** shows macrostate operations and **b** microstate operation. Closed boxes represent systems from which we only know some partial information, in this case the mean energy. Inside the box there is the actual microstate unknown to us if the box is closed. Scenario **a** shows the situation where one has an initial system of which only the mean energy $e$ is known and one can use any environment, being again limited to knowledge of its initial average energy $e_\beta$. The question is whether we can find a unitary $U$ that takes the two systems, regardless of what is actually inside of them, to one box for which we are certain that we will find inside the microstate $\rho_f$. The answer to this question is provided by scenario **b**, where the initial boxes of system and environment are both open (implying that we know what is the microstate) and populated with the maximum-entropy ensemble. $U$ exists if and only if there exists a unitary $U_{mic}$ that implements the transition in **b** when taking $\rho = \gamma_e(H)$. This shows that a thermodynamic transition is possible if and only if it is also possible under the assignment of ensembles to systems

if for any $\epsilon > 0$ and $\epsilon' > 0$ there exists an environment—that is, a set of $N$ systems with Hamiltonians $H_{E^i}$—and a unitary $U$ on SE, so that

$$\rho_f \approx_\epsilon \mathrm{tr}_E\left( U\rho \overset{N}{\underset{i=1}{\otimes}} \gamma_\beta(H_{E^i}) U^\dagger \right) \tag{11}$$

while preserving the overall mean energy

$$\mathcal{E}\left( U\rho \overset{N}{\underset{i=1}{\otimes}} \gamma_\beta(H_{E^i}) U^\dagger \right) \approx_{\epsilon'} \mathcal{E}\left( \rho \overset{N}{\underset{i=1}{\otimes}} \gamma_\beta(H_{E^i}) \right). \tag{12}$$

An operationally inspired illustration of the two types of operations as well as of our result is provided in Fig. 1.

In this setup, we call a macrostate $(e, H)$ and a microstate $\rho$ operationally equivalent, denoted as $(e, H) \sim_\beta \rho$, if

$$(e, H) \overset{\beta\text{-mac}}{\longrightarrow} \rho_f \Leftrightarrow \rho \overset{\beta\text{-mic}}{\longrightarrow} \rho_f. \tag{13}$$

Whenever a macrostate and a microstate are related by the equivalence $\sim_\beta$, then, concerning the possible thermodynamic transitions, they are equivalent descriptions of the system. We are now in a position to state our main result.

**Theorem 3:** (Equivalence with the canonical ensemble) For any $\beta \neq 0$, the macrostate $(e, H)$ is operationally equivalent to the corresponding canonical ensemble compatible with the partial information $e$. That is,

$$(e, H) \sim_\beta \gamma_e(H). \tag{14}$$

This theorem shows that, whenever the behaviour of a system under partial information concerns the possible thermodynamic transitions, a macrostate can be treated as if it was in its corresponding canonical ensemble, in the sense that they their behaviours coincide exactly. A sketch of the proof, for illustration of the idea, is given in Fig. 2. The full proof appears in the Supplementary Methods 1.

Lastly, let us note that all of the above, including the operations and the notion of operational equivalence, can straightforwardly be generalised to the more general case of a set $\mathcal{Q} = \{Q^j\}$ of $n$ commuting observables replacing $H$, a vector $\mathbf{v}$ of expectation values for each observable replacing $e$ and by now parametrising the environment by a vector of inverse 'temperatures' $\beta = (\beta^1, \ldots, \beta^n)$ encoding other intensive quantities. In this case, we obtain an operational equivalence between the macrostate $(\mathbf{v}, \mathcal{Q})$ and the corresponding maximum-entropy ensemble compatible with the partial information. More precisely, we obtain that, as long as

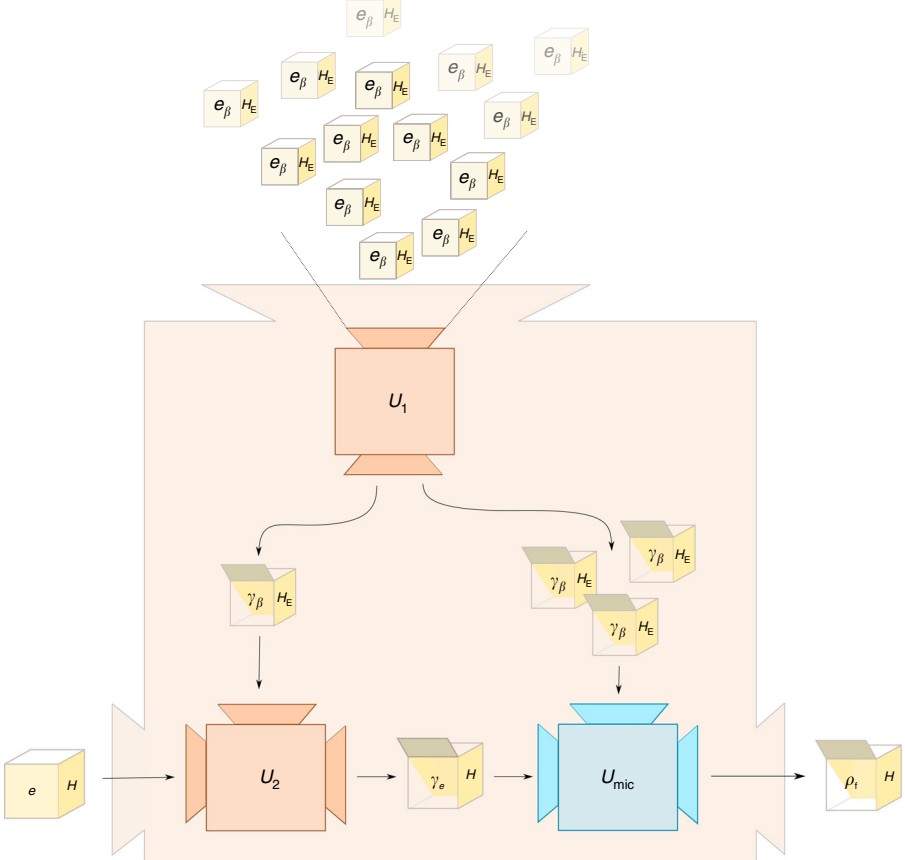

**Fig. 2** Sketch of proof of the main result. We show how an operation of the form of Fig. 1b can be used to build an operation of the form Fig. 1a. This gives the direction $\Leftarrow$ in (13) for the equivalence of Theorem 3 (the other direction is trivial, see Supplementary Methods 1). The construction has three sub-blocks: Box $U_1$ represents the fact that one can obtain the microstate $\gamma_\beta(H_E)$ to arbitrary precision from many copies of the macrostate $(e_\beta(H_E), H_E)$ using a macrostate operation (interestingly, this can be done with exact energy conservation). This result relies on a central limit theorem and typicality results for individual energy eigenspaces of many non-interacting systems. Box $U_2$ operates by choosing as $H_E$ as a rescaled version of $H$ and showing that one can then obtain the microstate $\gamma_e(H)$ using a macrostate operation. Box $U_{\mathrm{mic}}$ exists by assumption: it uses the microstate operation to obtain $\rho_f$ from $\gamma_e(H)$ (it is the one represented in Fig. 1b))

$\beta^j \neq 0$ for all $j$,

$$(\mathbf{v}, \mathcal{Q}) \sim_\beta \gamma_\mathbf{v}(\mathcal{Q}), \tag{15}$$

where, in exact analogy to (1), $\gamma_\mathbf{v}(\mathcal{Q})$ is the so-called generalised Gibbs ensemble (GGE)[11,19–23]

$$\gamma_\mathbf{v}(\mathcal{Q}) := \frac{e^{-\sum_j \beta_S^j(\mathbf{v}) Q^j}}{\mathrm{tr}\left( e^{-\sum_j \beta_S^j(\mathbf{v}) Q^j} \right)}, \tag{16}$$

with $\beta_S^j(\mathbf{v})$ being functions such that $\mathrm{tr}(Q^j \gamma_\mathbf{v}(\mathcal{Q})) = v^j$. The scenario and derivation is completely analogous to that yielding Theorem 3 and it is presented in the Supplementary Methods 1.

**Rederiving bounds on work extraction.** At a conceptual level, we regard as our main contribution the theoretical justification, from an operational perspective, for the common and empirically extraordinarily well-supported use of the canonical ensembles in thermodynamics to describe systems in settings of partial information. The key step in this justification has been to prove a coincidence in behaviour with respect to thermodynamic transitions. The relevance of this coincidence is that many thermodynamic tasks and the laws of thermodynamics can ultimately be formulated as reflecting state transitions.

We illustrate this first using the task of work extraction and then derive the second law of thermodynamics in the form of the Clausius inequality.

Let us consider the following task: One is given a system S from which only the Hamiltonian and its mean energy $e$ are given. For instance, S might be a burning fuel which one wants to use in a heat engine to perform work together with an environment. This common scenario is tackled in phenomenological thermodynamics by assigning to the system a temperature $T_S$ and to the environment a temperature $T$. The optimal amount of work that can be performed is simply given by the difference of free energies of S during the process. Note that phenomenological thermodynamics operates at a level where only partial information—the thermodynamic variables—are given about both the system and the environment. Furthermore, the operation of such a heat engine is effectively independent of the precise microstate that describes S and E, exactly in the same spirit as that of Definition 1.

From the perspective of statistical mechanics, the assignment of a temperature $T_S$ and $T$ is understood as the assumption that both systems are in a canonical ensemble. Indeed, if we assume the system and the environment are initially in the state

$$\gamma_e \otimes \gamma_\beta := \gamma_e(H) \otimes \gamma_\beta(H_E) \tag{17}$$

one can formally derive limitations on the work $\Delta W$. The problem amounts to finding how much one can reduce the energy of the whole compound by any unitary operation that does not conserve the energy and assuming that all of the remaining energy can be extracted as work. One then obtains that this value is determined by the free energy as (see, e.g., ref. [16])

$$\Delta W^{\mathrm{opt}} := \max_{U, H_E} \left[ \mathcal{E}\left( \gamma_e \otimes \gamma_\beta \right) - \mathcal{E}\left( U \gamma_e \otimes \gamma_\beta U^\dagger \right) \right]$$
$$= \Delta \mathcal{E}_S - T \Delta \mathcal{S}_S := \Delta \mathcal{F}_S, \tag{18}$$

where we denote the energy by $\mathcal{E}(\rho_{SE}) = \mathrm{tr}(\rho_{SE}(H_S + H_E))$, $\Delta \mathcal{E}_S$ is the energy difference on S and $\Delta \mathcal{S}_S$ is the difference of the von Neumann entropy on S. This yields the bound in terms of the free energy $\mathcal{F}_S = \mathcal{E}_S - \beta^{-1} \mathcal{S}_S$ of the system and it relies only on the

first law of thermodynamics $\Delta \mathcal{E}_{SE} = -\Delta W$ and the prescription of canonical ensembles to the system and environment.

We will now show that one can use Theorem 3 to derive the bound (18) without relying on the assumption (17) which assigns maximum-entropy ensembles to the systems at hand. The system S, given the partial information, is described by the macrostate $(e, H)$. We also have at our disposal an environment in any macrostate of the form $\otimes (e_\beta(H_{E^i}), H_{E^i})$. The goal is to perform work with a protocol in such a way that it achieves this work extraction for all possible microstates in the respective equivalence classes, $[e]_H$ and $[e_\beta(H_{E^i})]_{H_{E^i}}$ for all $i$, in a similar way to the way the laws of phenomenological thermodynamics allow one to extract work regardless of the actual microstates of the systems involved. It is clear that

$$\gamma_e(H) \overset{\beta\text{-mic}}{\longrightarrow} \gamma_e(H) \forall e, H. \tag{19}$$

Hence, by invoking Theorem 3 one has also that

$$\begin{aligned} (e, H) &\overset{\beta\text{-mac}}{\longrightarrow} \gamma_e(H), \\ (e_\beta(H_E), H_E) &\overset{\beta\text{-mac}}{\longrightarrow} \gamma_\beta(H). \end{aligned} \tag{20}$$

Once we have the system S and the environment E in the states of at the r.h.s. of (20), we simply apply the unitary achieving the maximum in Eq. (18). In this way an amount of work given by $\Delta \mathcal{F}_S$ is extracted. The fact that this is the optimal possible value that works for all microstates in $[e]_H$ is trivial, since the work extraction has to be successfully implemented if the system is given is in the state $\gamma_e(H) \in [e]_H$, for which the optimal value is $\Delta \mathcal{F}_S$ as given by Eq. (18).

We conclude then that the optimal work that can be extracted from a system and an environment, from which we only know their mean energy coincides precisely with the optimal work when system and environment are described by their corresponding canonical ensemble. A completely analogous argument applies to any other conceivable task that can be formulated as concerning state transitions between microstates, both thermodynamically but also, and more generally, tasks with other conserved quantities.

**Second law and Clausius inequality.** Now we show that the second law of thermodynamics can be recovered by using Theorem 3. More particularly, we show that the set of achievable states $\rho_f$ that can be reached by a transition of the form

$$(e, H) \overset{\beta\text{-mac}}{\longrightarrow} \rho_f \tag{21}$$

can be determined only by merely taking into account the free energy $\mathcal{F}$. First note that by Theorem 3 the set of achievable $\rho_f$ coincides with those that can be achieved by microstate operations of the form

$$\gamma_e(H) \overset{\beta\text{-mic}}{\longrightarrow} \rho_f. \tag{22}$$

The set of achievable states by microstate operations has been investigated in ref. [16], where it is shown that the transition is possible if and only if the free energy decreases. Hence, we arrive at the second law of the form

$$(e, H) \overset{\beta\text{-mac}}{\longrightarrow} \rho_f \Leftrightarrow \mathcal{F}(\gamma_e(H)) \geq \mathcal{F}(\rho_f). \tag{23}$$

Importantly, this result can also be seen as a derivation of the free energy as a state function $F(e, H)$ on macrostates, by setting $F(e, H) = \mathcal{F}(\gamma_e(H))$. Since the energy is already naturally defined for macrostates we then also obtain the derived Gibbs entropy

$$S(e, H) := \frac{1}{T}(e - F(e, H)). \tag{24}$$

Interpreting the change of energy on the system as heat $\Delta Q := e' - e$, we see that a transition $(e, H) \xrightarrow{\beta\text{-mac}} (e', H)$ between macrostates using macrostate operations is possible if and only if

$$\Delta Q \leq T\Delta S, \tag{25}$$

with $\Delta S := S(e', H) - S(e, H)$. We thus find that a state transition between macrostates is possible if and only if the Clausius inequality is fulfilled.

Lastly, we highlight that a generalisation of the same results for the case of multiple commuting observables is possible combining in a similar fashion Theorem 3 from the Supplementary Methods 1 with the results of ref. [17] to arrive at a formulation of the second law of the form

$$(\mathbf{v}, \mathcal{Q}) \xrightarrow{\beta-\text{mac}} \rho_f \Leftrightarrow \mathcal{G}(\gamma_\mathbf{v}(\mathcal{Q})) \geq \mathcal{G}(\rho_f), \tag{26}$$

where $\mathcal{G}$ is the so-called free entropy defined as

$$\mathcal{G}(\rho) = \sum_j \beta_j \text{tr}(\rho Q^j) - \mathcal{S}(\rho). \tag{27}$$

**Operational equivalence breaks for exact energy conservation.** Theorem 3 establishes the operational equivalence between macrostates and their corresponding maximum-entropy ensembles based, among others, on Assuption 4, where it is assumed that the mean value of the energy is preserved. In this section, we consider a strengthening of macrostate and microstate operations in which Assumption 4 is replaced by the following:

Assumption 5: (Exact energy conservation) The unitary evolution $U$ commutes with the total Hamiltonian,

$$[U, H_S + H_E] = 0. \tag{28}$$

We define, in full equivalence to the previous discussion, macrostate and microstate operations, but with exact preservation of the energy. We say that $\rho_f$ can be reached by commuting macrostate operations from the macrostate $(e, H)$, similarly to Definition 1, but imposing, instead of mean energy conservation as in Eq. (9), the condition (28). One can define, analogously, commuting microstate operations by imposing similarly Eq. (28) and a notion of operational equivalence $\overset{c}{\sim}$ analogous to (13).

In the Supplementary Methods 2, we show that for every $\beta$ and non-trivial $H$, there exists at least one initial value $e$, such that

$$(e, H) \overset{c}{\underset{\beta}{\sim}} \gamma_e(H). \tag{29}$$

We believe the proof of this result to be interesting in its own right, because in it we show that the maps produced by commuting macrostate operations admit a simple linear characterisation, the details of which are discussed in the Supplementary Methods 2. We leave as a relevant open question to investigate particular cases where equivalence with the maximum-entropy ensemble is recovered for exact energy conservation. In the Supplementary Methods 3, we present one

setting in which the operational equivalence for the commuting case is recovered locally for large non-interacting systems. Another possibly fruitful direction is to impose extra restrictions on the set of possible states within an equivalence class and show equivalence only for this restricted class. Some partial results on this question are discussed in the Methods section. Also, note that (29) holds even if one replaces Assumption 1 for the assumption that the bath is already in a canonical ensemble. This follows since a bath fulfilling Assumption 1 can be transformed into a canonical state by unitaries that respect (28) (see Supplementary Methods 1). Again, an analogous breakdown of the equivalence as given by (29) exists for several commuting observables.

With respect to the justification of the use of maximum-entropy ensembles, this result implies that one cannot justify, in general, assigning a maximum-entropy state to a system under partial information by means of considering the possible thermodynamic transitions in a setting of exact energy conservation. This, we submit, again confirms current practice, because canonical ensembles are rarely used in situations where full control is had over the microdynamics of a system. From an operational point of view, note that Eq. (28) can be interpreted as system and bath being isolated from any other external system during the transition. However, in a situation where the only information available about the external system is also its mean energy, it seems challenging to certify that indeed system and bath evolved truly isolated. In this case, one can only be certain that the external system did not change its mean energy, which gives rise to the weaker condition of Assumption 4. Nonetheless, we regard both mean energy or exact energy conservation as reasonable assumptions whose adequacy will depend on the particular description of the situation at hand.

**The macroscopic limit.** In the light of the inequivalence of macrostates and their respective ensembles for the case of exact commutation, it is interesting to quantify by how much one has to violate (28) in order to recover equivalence. For this, let us introduce the random variable $X$ which quantifies the energy change of SE during a macrostate operation. This energy change is captured by a probability distribution $P$. Theorem 3 implies the equivalence between the macrostate $(e, H)$ and its corresponding ensemble with macrostate operations. These preserve the mean energy of the compound, hence with vanishing value of the first moment of $P$, although higher moments could well be different from zero. On the other hand, in the case of commuting macrostate operations, all the higher moments of $P$ would indeed vanish due to condition (28). Hence, the deviation from zero of the higher moments of $P$ seems a sensible quantifier of the violation of (28).

We will now discuss the behaviour of these higher moments for large, non-interacting and independent systems, capturing the classical limit of macroscopic systems. To do so, consider a system S described by $N$ non-interacting subsystems. We will consider macrostate operations between a macrostate $(e, H)$ and a final state $\rho_f$ and impose that the final and initial states are large and uncorrelated. That is, instead of being any microstate in $[e]_H$, the initial microstate takes the form $\sigma = \otimes_{i=1}^N \sigma^i$. We also assume that the final state takes a similar form $\rho_f = \otimes_i^N \rho_f^i$. Using standard arguments of central limit theorems one can show that, in the limit of large $N$ and for bounded Hamiltonians, $P(X)$ for the transition $(e, H) \xrightarrow{\text{mac}} \rho_f$ converges in distribution to a normal distribution with variance scaling as $\sqrt{N}$. Hence, the higher moments of $P(X)$ per particle vanish (see Supplementary Methods 3). This is an argument in favour of the assignment of the ensemble to macrostates, for large weakly correlated systems, as long as one tolerates violations of (28)—as measured by the

higher moments—that are negligible in comparison with the typical energy scales involved in the thermodynamic operation.

**Comparison with existing work**. There exist several complementary approaches to justify the use of or single out maximum-entropy states in thermodynamics. As stated already in the introduction, the novelty of our approach lies in specifically assigning ensembles based on the set of possible thermodynamic transitions. This is in contrast with previous approaches, where canonical ensembles are justified based on measurement statistics of relevant observables. Both perspectives—the one presented here and previous approaches—can be fairly incorporated in a more general formulation about what is meant by a justification of the use of ensembles: the representation of a system's state by a statistical ensemble is justified with respect to some property if one can, on reasonable grounds, derive that the ensemble and the state behave exactly the same with respect to this property. Approaches based on notions of typicality usually consider as system states pure quantum states and the measurement statistics of some restricted set of observables—often local observables—as the property to be reproduced by the ensembles[4,6]. In contrast, in the present work, the system states are macrostates of partial information and the property is with respect to achievable state transitions under thermodynamic evolution. Theorem 3 justifies the assignment of maximum-entropy ensembles to macrostates with respect to such transitions. Macrostates are arguably the most common state assignment in thermodynamics, being at the root of discusssions of the link of statistical mechanics and phenomenological thermodynamics, in that one often has knowledge of a system's state only up to its expectation values. Hence, this result provides a very broad operational justification of the use of maximum-entropy ensembles for a plethora of thermodynamical processes.

Another aspect that distinguishes our approach from other notions based on typicality is that we do not need to introduce a measure on quantum states or make any particular assumption on the dynamics. More precisely, known approaches based on typicality consider a given subset of quantum states and show that measurement statistics coincide with those of the ensemble for most of the quantum states within the subset. However, there is no general argument to advocate that one will find in nature precisely those states for which the statistics resemble those of the ensemble, even though these states comprise the vast majority according to reasonable measures. In contrast, one of the main features of our results is that it works for all and not for most of the quantum states that are compatible with the partial information. First, we demand that the transitions from macrostates, as given abstractly by (3), reach $\rho_f$ for all the states compatible with the partial information. It would be analogous to the notion of typicality if we would instead demand that $\rho_f$ is reached only from most of the microstates according to some state measure, but this is actually not required to derive our main results. Secondly, the equivalence between the macrostate and the corresponding ensemble holds for all possible macrostates, instead of just for a vast majority of the macrostate according to some measure on the possible values of the partial information. Most importantly, we stress that the equivalence between the macrostate and the ensemble holds irrespectively of the system's dimension. To put it in more practical terms, our results imply that a system, even if made of a few qubits, behaves as if it was in its maximum-entropy ensemble when it comes to state transitions under joint evolution with a possibly large bath. This is true in a single-shot regime—considering transitions on a single copy of the system at hand—without having to rely on taking the thermodynamic limit where transitions of large number of copies are considered instead[24,25].

Lastly, it may seem that our approach is closely related to that of the famous Jaynes' principle according to which a system should always be assigned the maximum-entropy state consistent with what one knows about it[19,26]. What both approaches have in common is that they consider the question of assigning microstates to macrostates. However, apart from this they differ considerably: Jaynes motivates his principle on the basis of Shannon's findings about the uniqueness of the Shannon entropy as an asymptotic measure of information. In contrast, our approach does not require us to assume any privileged measure of information, or even rely on any consideration about information measures at all. Moreover, as noted in the preceding paragraph, our approach also makes no reference to an asymptotic setting. Instead, in our work, we define a task on an individual system and investigate how an experimenter's partial knowledge about the system impacts her ability to execute this task. The canonical ensemble then naturally emerges as an effective representation of the experimenter's operational abilities in this setting. Again, no recourse to a measure of information, average performance, or even a subjectivist account of probabilities is required in our setting.

## Discussion

In this work, we have introduced a fresh way of justifying the very common use of maximum-entropy ensembles as a representation of the state of systems. We take a strictly operational stance to the subject, in which an experimenter has only partial information about the microstate of the system and all operations have to be compatible with such partial information. The vantage point for our argument concerns the possible thermodynamic transitions that systems can possibly undergo. This approach has the key advantages that it (a) naturally fits with many operational tasks in thermodynamics and its laws and (b) does not require underlying typicality arguments, and hence avoids some of their conceptual issues. We have also shown how our results can be used to derive features of phenomenological thermodynamics, such as the Gibbs entropy, free energy as state functions and the Clausius inequality, which determines whether a state transition on macrostates is possible without investing non-equilibrium resources. We are thus able to derive fundamental thermodynamic results without any assumption about typicality or information measures. Finally, our results generalise to the setting of several commuting observables. As such, the results here are likely to be of interest for thermodynamics in generalised settings or even outside the context of thermodynamics.

We point out as interesting further direction of research to incorporate probabilistic transitions to our formalism. We assume in our formalism that macroscopic operations transform any state of the equivalence class into a desired final state. It is an interesting endeavour to consider possible relaxations of this requirement by allowing some error probability on the transition. We leave it as an open question to investigate sets of reachable states under such relaxations. Lastly, the findings that operational equivalence breaks down for exact commutation suggest that further investigation is needed. In particular, it is natural to ask if one can impose additional constraints or assumptions to recover equivalence under exact energy conservation.

**Data availability**. Data sharing not applicable to this article as no data sets were generated or analysed during the current study.

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

## Acknowledgements

We thank H. Tasaki for comments. This work has been supported by the ERC (TAQ), the DFG (GA 2184/2-1, CRC 183, B02), the Studienstiftung des Deutschen Volkes, the EU (AQuS), and the COST action MP1209 on quantum thermodynamics.

## Author contributions

P.B., H.W., J.E., and R.G. conceived the research question and wrote the article. P.B., H.W., and R.G. derived the technical results.

## Additional information

**Competing interests:** The authors declare no competing financial interests.

