## [Peer Review File · Nature Communications]

REVIEWERS' COMMENTS:

Reviewer #1 (Remarks to the Author):

This paper addresses thermodynamic operations and compatibility between partial information (e,H) and maximally-entropy ensemble. Suppose we know only partial information such as energies and Hamiltonians (e,H) . Then the question is the condition to find Unitary operator leading to the final state ρ_f in the system, 'regardless of the details on the micro states'. The authors found that this is possible if and only if unitary transformation exists, which connects maximally-entropy ensemble to the final state ρ_f . This is consistent with the usual belief in the thermodynamic operation, provides a strong basis on it. I think this question and approach is new and very important in the quantum thermodynamics. So in my opinion, it is worth while be published.

Comments and questions are the following for the publication.

1) Is there limitation on the final state ρ_f ? I do not guess that one cannot get arbitrary ρ_f . If so the authors should explain.

2) I believe that the approach is completely new. But I want to check the following. Any derivation of canonical ensemble, time-evolution is considered, where the steady state is described by the diagonal ensemble. People discuss that the diagonal ensemble is consistent with the canonical ensemble on observables in subsystems. This argument is usually done for an isolated system. Difference between this and the present paper is that the present paper includes thermal reservoirs. What is a crucial role of thermal reservoir ?

3) Details of U are not shown here since the authors care only on the existence. Is there nontrivial example which changes the system Hamiltonian in time ?

Reviewer #2 (Remarks to the Author):

The authors study a problem of fundamental importance to thermodynamics: The emergence of the canonical ensemble from operational principles. The canonical, or Gibbs-, ensemble is the foundational building block of thermodynamics as it is inevitably used in thermodynamic reasoning, from microscopic derivations of thermodynamic laws in resource theories to deriving practical limits for the performance of machines. In the past, there have been various approaches that can be loosely summarised by two paradigms: a) how does a system, which I have full microscopic information about (i.e. a pure state), behave like a maximum entropy state dynamically or for reasonable observables and b) Given partial information, what can I practically achieve without adding unjustified assumptions. The authors follow the second approach, which corresponds to an operationally minded view on thermodynamics: Given some partial information (Hamiltonian and its expectation value) and an environment with partial information, which final states can I always reach given average energy preserving global unitary operations. The fact that this set is not only not empty, but corresponds exactly to the set of final states reachable by canonical ensembles compatible with this partial information, is one of the most profound insights into the problem I have seen in years. I have some minor questions and suggestions:

1) From a philosophical point of view, I find it amazing that the final states are reachable from all micro-compatible states with a set of expectation values is really equal to the canonically reachable set of states. But of course, for any given physical situation, the microstate may in fact be different and this can be witnessed by final states having different properties than expected from the canonical predictions. One could in turn ask: how likely is it that a given operation takes my target system outside of the set of canonically reachable states. I guess this gives rise to a set of questions similar to canonical typicality or dynamical equilibration, is that correct?

2) I am a bit puzzled by the average vs. exact energy preservation. I've always understood exact energy preservation (i.e. generator of the time translation commuting with itself) as a basic axiom of quantum mechanics. Relaxing this to only average energy conservation implies some sort of work cost for implementing these operations (i.e. a fluctuation storage device that is external to the system and environment considered). And of course one also needs a clock to precisely time the switching of the interaction hamiltonians inducing the unitary operations. So while this implies that the description is not entirely self-contained it is really surprising that this is actually necessary for the theorem to hold. The argument in appendix D is really elegant, but it leaves open the most important question for me: can one find a set of states for which the cases do indeed coincide? And what about the case of Gibbs environment with partial information about the system? At least in this case I would have suspected equivalence in some asymptotic fashion. I think this point is really important and could benefit from an extended discussion. Also the sentence "More- over, note that from an operational point of view the setting of commuting macrostate operations appears unnatural, because in it one assumes that an experimenter has no access to the microstate information at the level of the systems, while having full microstate level control over the operations that she implements."- I don't immediately see this implication. Commuting macrostate operations just imply a self-contained description, I think this sentence needs some further clarification. Or is it self-evident and I missed something?

While the above two points merit a bit of an extended discussion in my opinion, I would nonetheless strongly recommend publication of this article. It is a genuinely novel way of arriving at the canonical ensemble, which is one of the central problems of statistical mechanics and thermodynamics, and I have no doubt that it will inspire further research in this direction.

Reviewer #1 (Remarks to the Author):

We would like to thank the referee for this very careful, helpful and at the same time positive report. We are obviously pleased that the referee considers our approach "completely new" and very important in quantum thermodynamics. Some important points have been raised by the referee to which we respond in detail below. In this reply letter, we detail what changes we have made to fully accommodate the remarks.

1) Is there limitation on the final state ρ_f ? I do not guess that one cannot get arbitrary ρ_f . If so the authors should explain.

This is a good point. Yes, one can characterize the limitations on the reachable ρ_f . Using well-known results in quantum thermodynamics one can show that the reachable ρ_f are the ones that have less free-energy than the corresponding maximum-entropy ensemble. This is now given in Eq. (23) in the current version.

2) I believe that the approach is completely new. But I want to check the following. Any derivation of canonical ensemble, time-evolution is considered, where the steady state is described by the diagonal ensemble. People discuss that the diagonal ensemble is consistent with the canonical ensemble on observables in subsystems. This argument is usually done for an isolated system. Difference between this and the present paper is that the present paper includes thermal reservoirs. What is a crucial role of thermal reservoir ?

We thank the referee for the enthusiastic assessment. Thm 3. implies, in particular, that all states in an equivalence class can be transformed, by a unique operation, into the canonical ensemble. Since the canonical ensemble is the maximum-entropy state of the system, one needs to have an external system, a bath, that provides that entropy. This is a possible way of explaining the role of the bath, which is very appropriate for comparison with the other approach that the referee mentions.

As the referee correctly points out, time-evolution can drive isolated systems, under some conditions and assumptions, in such a way that the statistics of certain observables are stationary and similar to those of the canonical ensemble. It is important here to point out that this emergence of the canonical ensemble also requires the use of a reservoir. This is because the isolated system output statistics compatible with the canonical ensemble only when very coarse-grained observables are measured. That is the case when the system is measured only in small subsystems or when measuring average values per particle. In those cases, one is effectively tracing out a huge portion of the total system and focusing only on the small subset of degrees of freedom that are relevant for the measurement. Only for those degrees of freedom is the system behaving like a canonical ensemble while the remaining degrees of freedom play the role of a reservoir providing entropy. Hence, although equilibration to a canonical ensemble is said to occur for isolated systems, implicitly it also uses the notion of a reservoir or bath in its construction. From this perspective, both this approach and ours are similar. We would like to emphasize, however, that while such thermalization behaviour in closed systems is expected in many situations, a rigorous proof of it for large isolated systems remains one of the outstanding problems in theoretical many-body physics. In particular there are classes of interacting many-body models, such as many-body localized models, where this behaviour does indeed not appear.

3) Details of U are not shown here since the authors care only on the existence. Is there nontrivial example which changes the system Hamiltonian in time ?

Indeed, we consider the main result the existence of such unitary. But our proof is constructive and we do provide the specific form of the unitary in the supplementary material. To summarize, it is a three-step unitary evolution U including:

- 1.- Mixing over all the eigenspaces of the heat bath.
- 2.- Tracing out all of the bath, apart from a subsystem.
- 3.- Performing a swap between this remaining subsystem and the system S on which the transition is performed.

This 3-step unitary U can of course –as any unitary– in principle be realised simply by letting the system evolve under a Hamiltonian proportional to $\log(U)$.

Nonetheless, the example above is only one example of a unitary performing the desired transition, but the existence of other unitaries with the same effect is to be expected. We leave as an interesting open question to find simplified implementations that operate equivalently, if not with full generality, in restricted cases of interest. Again, we would like to thank the referee for the interesting and very positive report. Having accommodated all remarks, we hope that in its present form, the manuscript is suitable for publication.

Reviewer #2 (Remarks to the Author):

We would like to thank the referee for the enthusiastically positive and careful review. We also thank him or her very much for the interesting questions raised and suggestions made. We have taken them very seriously and have altered the manuscript accordingly to accommodate them. We reply to them individually:

1) From a philosophical point of view, I find it amazing that the final states are reachable from all micro-compatible states with a set of expectation values is really equal to the canonically reachable set of states. But of course, for any given physical situation, the microstate may in fact be different and this can be witnessed by final states having different properties than expected from the canonical predictions. One could in turn ask: how likely is it that a given operation takes my target system outside of the set of canonically reachable states. I guess this gives rise to a set of questions similar to canonical typicality or dynamical equilibration, is that correct?

This is a good point. First, note that in order to be able to say something about the probability of reaching a state outside of the set of canonically reachable states, one would need to assume a given initial probability distribution for the states compatible with the partial information. That is, one can ask, given a probability measure for states with a given mean energy, which states can be reached when allowing for a given error in the transition? We leave this approach to the problem as an open question for further work. We have mentioned it explicitly in the final discussion.

A very related, but similar problem is to ask whether a randomly chosen energy-preserving unitary (on system and bath) would map some microstate in a given equivalence class outside of the set of canonically reachable states.

In this case, we believe that this probability (given some reasonable measure on the set of energy-preserving unitaries) is indeed extremely small, since with very high probability (for a large bath) the system should simply thermalize, i.e., be mapped to the canonical state at the bath's temperature. In this sense we share the intuition that conditions similar to canonical typicality should arise. We emphasize however, that such an operation would also correspond to one where all the non-equilibrium free energy of the system is dissipated into the environment, so that the system would be useless for further thermodynamic operations such as work extraction. We find these questions very relevant, but we feel that on a rigorous level our answers are yet too speculative to include them in the current manuscript.

2) I am a bit puzzled by the average vs. exact energy preservation. I've always understood exact energy preservation (i.e. generator of the time translation commuting with itself) as a basic axiom of quantum mechanics. Relaxing this to only average energy conservation implies some sense of work cost for implementing these operations (i.e. a fluctuation storage device that is external to the system and environment considered). And of course one also needs a clock to precisely time the switching of the interaction hamiltonians inducing the unitary operations. So while this implies that the description is not entirely self-contained it is really surprising that this is actually necessary for the theorem to hold. The argument in appendix D is really elegant, but it leaves open the most important question for me: can one find a set of states for which the cases do indeed coincide?

We also find very relevant the question on the existence of sets of sets for which equivalence is restored. More formally, the question is: are there subsets of states within an equivalence class, so that for those states only, the set of reachable states coincides with the canonical ensemble (for exact energy preserving operations)?

We cannot answer in full generality, but we can answer it in the negative for a very relevant class of subsets of states. This comes from the fact that the proof of appendix D (in the old version of the manuscript, now Supplementary Methods 2), where the breakdown of equivalence is shown, follows as long as the subset of states spans a vector space with the same dimensionality as the whole set of states in the equivalence class. For instance, take as a subset of states those inside the full equivalence class which are within an epsilon-ball (in a given norm) around a fixed one. For this subset, no matter how small epsilon is, we find as well a breakdown of the equivalence with the canonical ensemble. We see this as an indicator of the difficulty of finding subsets so that equivalence is recovered. But we leave it as an open question in full generality. We have included a comment in the main text and in the supplementary methods mentioning what we explained above.

And what about the case of Gibbs environment with partial information about the system? At least in this case I would have suspected equivalence in some asymptotic fashion. I think this point is really important and could benefit from an extended discussion.

The breakdown of the equivalence still holds if one assumes a Gibbs environment. The reason is that it is possible to transform an environment from which only partial information is known to an ensemble in a Gibbs state by using only exact energy conserving unitaries. Hence, having a Gibbs environment does not alter the reachable states.

However, it is true that in a certain asymptotic limit "local equivalence" can be obtained also in the case of exact energy preservation and we have added a discussion on this in the supplementary methods, together with a comment in the main text. We therefore only discuss it briefly here: Consider a large system made up of many uncorrelated and non-interacting parts. For simplicity simply assume the parts are all identical and described by the macrostate (e, H) . In the case of average energy-conservation, the full system is then operationally equivalent to the global Gibbs state of the many non-interacting systems. On the other hand, in the case of exact energy preservation it is possible to show that in the limit of N going to infinity, the N local subsystems, are all, individually, operationally equivalent to their local Gibbs states. However, it is not possible to guarantee operational equivalence between the global macrostate of the N subsystems with the global Gibbs state of the N systems as in the case of average energy preservation.

Also the sentence "Moreover, note that from an operational point of view the setting of commuting macrostate operations appears unnatural, because in it one assumes that an experimenter has no access to the microstate information at the level of the systems, while having full microstate level control over the operations that she implements."- I don't immediately see this implication. Commuting macrostate operations just imply a self-contained description, I think this sentence needs some further clarification. Or is it self-evident and I missed something?

Indeed, exact energy conservation would imply a self-contained description. What we mean with this sentence is only that the following situations

- i) the system-bath are perfectly isolated from the environment (self-contained),
- ii) a third system intervenes in the transition, although not pumping energy on average, and system-bath are not fully isolated,

might be indistinguishable if information of all systems (including the environment/third system) is only available in the form of mean energy. Then, since i) might be impossible to verify, one might be content with isolation on average ii). This is only meant as a possible operational justification of the assumption of mean energy conservation. We regard both scenarios as reasonable alternatives, which can be applied depending on the situation at hand.

We have extended the discussion on this point in the new version. Again, we thank the reviewer very much for the very positive and deeply thoughtful report. Having accommodated all comments, we hope that in its present form, the manuscript is suitable for publication.